# Deriving Exact Mathematical Models of Malware Based on Random Propagation

**Rodrigo Matos Carnier** [1,*] , **Yue Li** [2] , **Yasutaka Fujimoto** [2] and **Junji Shikata** [3]

1   Information Systems Architecture Research Division, National Institute of Informatics,
    2-1-2 Hitotsubashi, Chiyoda City, Tokyo 101-8430, Japan
2   Department of Electrical and Computer Engineering, Yokohama National University, 79-5 Tokiwadai,
    Hodogaya Ward, Yokohama 240-8501, Japan
3   Graduate School of Environment and Information Sciences, Yokohama National University, 79-5 Tokiwadai,
    Hodogaya Ward, Yokohama 240-8501, Japan
*   Correspondence: rodrigo.carnier@gmail.com

**Abstract:** The advent of the Internet of Things brought a new age of interconnected device functionality, ranging from personal devices and smart houses to industrial control systems. However, increased security risks have emerged in its wake, in particular self-replicating malware that exploits weak device security. Studies modeling malware epidemics aim to predict malware behavior in essential ways, usually assuming a number of simplifications, but they invariably simplify the single most important subdynamics of malware: random propagation. In our previous work, we derived and presented the first exact mathematical model of random propagation, defined as the subdynamics of propagation of a malware model. The propagation dynamics were derived for the SIS model in discrete form. In this work, we generalize the methodology of derivation and extend it to any Markov chain model of malware based on random propagation. We also propose a second method of derivation based on modifying the simplest form of the model and adjusting it for more complex models. We validated the two methodologies on three malware models, using simulations to confirm the exactness of the propagation dynamics. Stochastic errors of less than 0.2% were found in all simulations. In comparison, the standard nonlinear model of propagation (present in ∼95% of studies) has an average error of 5% and a maximum of 9.88% against simulations. Moreover, our model has a low mathematical trade-off of only two additional operations, being a proper substitute to the standard literature model whenever the dynamical equations are solved numerically.

**Keywords:** malware model; random propagation; Markov chain

**MSC:** 34F05

## 1. Introduction

Malware is one of the most ubiquitous cyberthreats of modernity. The Internet, as the natural environment of malware, creates an incredibly complex and ever-diversifying scenario for exploitation. As a consequence, this environment resembles more and more life ecosystems, with malware behaving (and evolving) similarly to diseases, while defense systems try to catch up with more robust measures like the immune system of our bodies. The advent of the Internet of Things deepened the similarities even further. Billions of Internet-connected devices with low specifications and limited security capability are being deployed every year [1,2], creating even more niches to be exploited. Malware specific to IoT readily emerged. As can be seen, studies on cybersecurity are more urgent than ever to lay the foundation of sustainable development of the Internet in the age of the Internet of Things.

Weak security measures implemented in IoT devices and networks are a chronic problem that is being continuously exploited by self-replicating malware. In recent years,

IoT botnets performed DDoS attacks that affected millions of users [3]. Mirai, the most successful of them, infected half a million devices and took down services like Netflix and Twitter for hours [2,4]. To hide the origin of Mirai, its creators uploaded the code to Internet forums, leading to a whole strain of Mirai "offsprings", each "evolving" with particularities to avoid the countermeasures to Mirai. Evolving threats effectively led to the creation of a field of malware taxonomy, with reports on diverse strains of traditional malware [5], malware for mobile devices [6], and malware specific for IoT [7].

With so many similarities, it is no surprise the scientific study of malware approached the field of biological epidemics. Since the seminal work of Kermack and McKendrick [8], epidemic models are developed as stages of disease progression, dividing the affected biological populations into demographics experiencing specific stages: susceptible to infection, incubating, infectious, in recovery, quarantined, immunized or deceased. All these biological concepts were then adapted or directly applied to "populations" of malware hosts, the network devices (see Table 1 for a summary of the notation used throughout this paper). While this is the most used baseline approach to modeling malware, over the years, many mathematical and simulation studies explored different modeling methodologies, but most of them still incorporate compartmental models.

**Table 1.** Definitions and symbols.

| Independent Variable | | States | |
| --- | --- | --- | --- |
| $T$ | Turn, where $T \in \mathbb{N}$ | $N$ | Total number of devices |
| | | $S$ | Susceptible devices |
| | | $E$ | Exposed devices |
| | | $I$ | Infected devices |
| | | $R$ | Recovered devices |
| | | $D$ | Dead devices |
| Common Parameters | | Special terms of this study | |
| $\alpha$ | Infection rate | $I_n[T]$ | Successful infections in a turn |
| $\beta$ | Detection rate | $M_n[T]$ | Cleaned infections in a turn |
| $\mu$ | Death rate by infections | $C[T]$ | Cost of mitigation (total reset devices) |
| $\tau$ | Infection delay | $I[T_f]$ | Value of convergence of I |
| $\phi$ | Recovery delay | $T_c$ | Time constant of I (95% of $I[T_f]$). |

Currently, the majority of studies develop models in the format of differential equations/Markov chains [9–13], agent-based models [14,15], cellular automata [16], or pure stochastic models based on simulations [17,18], but there are also hybrid models appearing [19,20]. A more detailed explanation and discussion of the state of the art will be presented in Section 2.

Markov chains based on compartmental models are the most popular approach in the literature. While reviewing studies on models of random propagation, it was noticed that all studies assumed a simple model of random propagation and focused their contribution on modeling different compartmental states and parameters to make the model more realistic. The most popular model used to describe random propagation (which will be called "standard nonlinear model" in this paper) is shared by more than 95% of the reviewed literature, consisting of a very simple expression of three terms: $\alpha S(t)I(t)$, where $\alpha$ is the rate of infection, $S$ is the number of susceptible devices, and $I$ is the number of infectious devices. While this model is accurate enough to approximate the dynamics of random propagation, it ignores one of two dynamics of inefficient target selection during random malware propagation, resulting in an overestimation of the speed of propagation. These dynamics were never explained in detail or explicitly in the literature before our work, nor an exact model of random malware propagation was ever developed to take them into account. In our previous work [21], we presented the first exact model of random propagation in discrete form, provided an in-depth analysis of these dynamics of inefficiency, and derived the exact Markov chain for the simplest cyclic epidemic model

SIS. This paper discusses in detail the modeling assumptions of random propagation and extends the derivation methodology to systems with more states. Step-by-step examples of our methodology are provided for three epidemic models of increasing complexity, as well as the principles to modify the propagation model and incorporate it into different malware models without repeating the burdensome deriving process, demonstrating the generality of methodology and proposed propagation model. The advantage of our model is the exact representation of random propagation, with a trade-off of only two additional mathematical operations compared to the standard nonlinear model, whose errors can reach 9.88% against simulations (the errors of our model are all below 0.2%). The disadvantage of our model is the discrete formulation and the dependency on Markov chain assumptions in the subdynamics of random propagation, i.e., to calculate the propagation of malware, the dynamics need to depend only on the previous state of the system. However, we note that this dependency does not apply to the complete malware model itself, as we will show by deriving the exact dynamics of propagation for the malware model SEIRS, which has a delay on two-state transitions and needs to keep track of past states beyond the last.

In summary, the contributions of this paper are:

1.  The generalization of the methodology of derivation of the dynamics of random propagation for any compartmental model of malware.
2.  The proposal of an alternative rule-based methodology of derivation, carried out by modification of the simplest form of the propagation dynamics.
3.  An analysis of the impact of the propagation model in complex malware models, including incubation, temporary immunity, and network heterogeneity.

The rest of this paper is organized as follows: Section 2 presents a review of the literature and related works. Section 3 makes an in-depth discussion of random propagation and its two mechanisms of inefficiency. It also presents our proposed canonical states of random propagation and defines the proper way to model random propagation as Markov chains. Section 4 showcases our methodology of the derivation of exact Markov chains of random propagation using three epidemic models, SIS, SEIRS, and 2SEIS. Section 5 presents the derived complete dynamics of the three epidemic models using our proposed propagation model. Section 6 presents the validation of our proposed model and compares its performance with the most popular model in the literature. Section 7 summarizes the paper and presents our conclusions.

## 2. Review of Literature Models

Malware models describe the rate of change of states in a network, in particular the states of devices with respect to behavior before and after infection. All devices presenting the same state are grouped in a population of the same name, whose size is tracked by the model. In this work, we will limit our analysis to discrete models, which are formulated as difference equations:

$$X[T+1] - X[T] = f(X[T]) \tag{1}$$

where $X[T]$ is the vector of states representing the size of different populations, $T$ is the current instance of discrete time, and $f(X[T])$ is the resulting dynamics of malware that alter the size of the populations.

The use of compartmental models of epidemics to represent the change of states in populations started with Kermack and Kendrick, who proposed an SIR model that described an epidemic as a noncyclic process of susceptible individuals being infected and then acquiring a recovered-immunized state. This simple yet insightful model introduced the two basic compartmental dynamics: the infection dynamic $\alpha IS$ and the recovery

dynamic $\beta I$, shown in Equations (2a)–(2c). Since then, additional dynamics were introduced in the form of additional states, transitions between states, and parameters.

$$
\begin{cases}
S[T+1] - S[T] = -\alpha S[T]\dfrac{I[T]}{N} & \text{(2a)} \\[2mm]
I[T+1] - I[T] = \alpha S[T]\dfrac{I[T]}{N} - \beta I[T] & \text{(2b)} \\[2mm]
R[T+1] - R[T] = \beta I[T] & \text{(2c)}
\end{cases}
$$

The most common states modeled for network populations are susceptible (S), exposed (E), infected (I), recovered (R), and dead (D). Susceptible devices operate normally. They are not infected yet and thus susceptible to infection; they are the equivalent of healthy human populations that were never infected by the pathological agent in question and do not have antibodies or any immunity developed to it. Exposed devices are already compromised by malware but not yet fully expressing the infection; they are equivalent to human individuals whose infection is in a phase of incubation. In the case of computer malware, this state usually represents infected devices that are not actively reproducing the malware yet. Infected devices are carrying the malware and actively reproducing it. Recovered devices were recently cleaned and acquired some form of immunity, be it to the single type of malware that previously infected it or to a group of malware; this immunity can be both permanent or temporary. Dead devices are usually not operating at all, either because malware took it down or because the very network administration answered to the epidemic by shutting it down indefinitely (meaning that the time it takes to perform the device maintenance is bigger than the time window taken by the epidemic to fully develop).

All malware mathematical models are based on a mix of these common states, developing specific transitions between them according to the modeled network. Examples can be found in SIRS models [22,23], SEIRS models [24–26], or SEIRD models [27,28]), to cite the most popular combinations. On top of them, innumerable variations introduce one or more new subdynamics, e.g., [22], where an SIRS-L model was proposed to account for the low-energy mode of wireless sensors.

However, modeling just stages of malware infection can lead to models that fail to minimally represent the dynamics of IoT malware epidemics, due to the increased (and increasing) complexity of IoT networks. For example, instead of static network structures connected to the Internet through a single gateway, IoT networks have many fluid structures that can connect smartphones and other mobile devices intermittently, making it harder to isolate and control access to the network [29]. Moreover, the diversity of software platforms on which Internet connectivity is built marks IoT networks as strongly heterogeneous compared to former networks [30], increasing surfaces of attack. This complexity can be introduced in models by adding more parameters and elements to the transition between states, like the model of [28] did to model heterogeneous IoT networks where malware spreads both via standard network infrastructure and device-to-device connections (Equations (3a)–(3e)):

$$
\begin{cases}
S[T+1] - S[T] = \mu_{bd} - \alpha_{1SE}S[T] - \alpha_{2SE} - \delta S[T] - \gamma S[T] + \lambda R[T] & \text{(3a)} \\[2mm]
E[T+1] - E[T] = \alpha_{1SE}S[T] + \alpha_{2SE}S[T] - \alpha_{1EI}E[T] - \alpha_{2EI}E[T] - \gamma E[T] - \delta E[T] & \text{(3b)} \\[2mm]
I[T+1] - I[T] = \alpha_{1EI}E[T] + \alpha_{2EI}E[T] - \gamma I[T] - (\delta + \delta_{ex})I[T] & \text{(3c)} \\[2mm]
R[T+1] - R[T] = \gamma(S[T] + E[T] + I[T]) - \delta_{RD}R[T] & \text{(3d)} \\[2mm]
D[T+1] - D[T] = -\mu_{bd}\delta(S[T] + E[T]) + \delta_{RD}R[T] + (\delta + \delta_{ex})I[T] & \text{(3e)}
\end{cases}
$$

where $\mu_{bd}$ is a constant birth rate and death rate of devices; $\gamma$ is the probability of successful patching, $\lambda$ is the probability of losing immunity; $\delta$ and $\delta_{ex}$ are, respectively, the probability of a node exhausting power either naturally or due to malware; $\delta_{RD}$ is the probability of natural death of recovered devices; $\alpha_{1SE}$ and $\alpha_{2SE}$ are, respectively, the probability of transition of nodes from state $S$ to $E$ either by infrastructure network communication or by device-to-device communication, and $\alpha_{1EI}$ and $\alpha_{2EI}$ are equivalent for $E$ to $I$.

But despite this main trend of research of proposing more realistic models by diversifying the modeled dynamics, the improvement of the main subdynamics of malware epidemics— the dynamics of random propagation of malware—has been largely ignored, remaining in the same simplified form since the Kermack and Kendrick model. This is our present focus: to improve the current dynamics of malware propagation into exact models of random propagation. To understand this research gap, we will review the modeling of random propagation separately.

*Random Propagation of Malware: A Subcomponent of Malware Models*

Although mathematical modeling of IoT malware theory has branched into important subfields and proposed complete models with valid choices of simplification, the dynamics of the propagation of malware is a subelement of modeling that has remained universally simplified since the seminal work of Kermack and Kendrick. Here, "propagation dynamics" means the logic of selection of targets used by malware as well as the mathematical expression that describes the rate of change of susceptible populations into compromised states (be it infected, exposed, or others). This submodel can be found in virtually every model in the form of $\alpha S(t)I(t)$. Excluding models that develop very specific dynamics of propagation, due to highly constrained possibilities of contact between individuals, all models in the literature assume global random propagation dynamics. (On a note, random propagation dynamics is also universally assumed because it can be easily modified to include simple constraints in the possibilities of contact between individuals, working as a canonical form of propagation).

From all studies that assume global random propagation, the overwhelming majority use the Kermack and Kendrick submodel to represent it: according to our review, more than 95% of the literature. It is so widespread that it will be called the "standard nonlinear model" in the rest of this paper. The remaining 5% of models found in the literature use either the most simple model possible in the form of the linear expression $\alpha S(t)$, or do not represent random propagation at all. Below is a discussion of these two most common models for global random propagation of malware. They will be presented in discrete time to facilitate comparisons with our proposed discrete model later in the paper.

**Linear Model** [9,10]: In this model, randomness is abstracted as just the rate of infection $\alpha$. The number of infections is proportional to the population of susceptible devices, and no consideration is given to the number of attacking devices.

$$I_{n/lin}[T] = \alpha S[T] \tag{4}$$

where $I_{n/lin}[T]$ is the number of devices infected this turn only, according to the linear model. It is different from both the population of infected devices $I$ and its variation $\Delta I$ (the latter also depends on the number of newly cleaned devices this turn).

This model is very convenient to insert in complicated malware models because it yields closed-form solutions more easily, but the error of estimation of infected populations can reach 50% [21]. This is due to the lack of consideration to the number of infected devices that are performing attacks.

**Standard Nonlinear Model** [11–15]: In this model, randomness is modeled as the number of encounters between susceptible and infected devices [11]. The expression of the linear model is multiplied by the percentage of infected devices in the network, yielding Equation (5).

$$I_{n/std}[T] = \alpha S[T]\frac{I[T]}{N} \tag{5}$$

This model is more realistic because few infected devices will lead to few attempted infections during a turn, regardless of the availability of susceptible devices and the rate of infection. It also takes into consideration that infected devices can target another infected device (since the choice of target is random between all the network members). However, it does not consider the fact that random attacks are uncoordinated and can lead to devices attacking the same target. An important consequence of this is that the standard nonlinear model *overestimates* the number of infections. Refer to our previous work for a detailed consideration of the difference between the pseudo-random propagation present in this model and the true random propagation that considers both dynamics.

## 3. Random Propagation of Malware

### 3.1. Basic Definitions

Our proposed models are based on the following the assumptions:

1. Discrete model of time.
2. Fixed number of network devices.
3. Markov chain assumptions for propagation dynamics.
4. No local constraints to global propagation.

Discrete models will be used throughout this paper, both for the proposed models of propagation and the complete malware models that utilize them. Time is defined as discrete turns, changing incrementally from one $[T]$ to $[T+1]$. Moreover, the modeled network model has a fixed number of devices $N$ and is isolated from exterior contact. The propagation of malware is considered to start from an initial number of already-infected devices.

All propagation models will be derived as Markov chains, which only depend on the current state to define the next. As a side note, only the dynamics of malware propagation need to satisfy Markov chain assumptions, not the entire malware model. Examples will be shown in the second and third malware models studied in this paper. The last assumption involves the necessity of global reach and no bias towards target selection by attacking devices, i.e., no local constraints.

As long as these assumptions are satisfied, our model of random propagation and its methodology of derivation are applicable to any malware model that spreads randomly.

Three different systems of network + malware were studied. Each had a different combination of compartmental states: (1) the simplest compartmental epidemic model susceptible–infected–susceptible (SIS); (2) an extended model susceptible–exposed–infected–recovered–susceptible (SEIRS) with time delay dynamics; and (3) a heterogeneous network with different behavior regarding malware infection, which was modeled as a double susceptible–exposed–infected–susceptible model and named 2SEIS.

In order to derive the complete dynamics of an epidemic system, two terms are necessary: a propagation term, $I_n$, which represents the rate of transition of susceptible devices into a compromised state (which will vary depending on the compartmental model), and a mitigation term, $M_n$, which represents the rate of recovery of infected devices into a susceptible or recovering state, depending on the model. The term $I_n$ will vary along this paper, but for the sake of simplicity, the same mitigation term will be assumed for all models: a simple, standard form of mitigation with network-wide scans that detect infections with $\beta$ detection rate and clean the detected infections. Therefore, the term for this mitigation of malware device-by-device is given in Equation (6):

$$M_n[T] = \beta I[T] \tag{6}$$

Once $I_n$ is derived, it can be composed with $M_n$ to build the vector $\Delta X[T]$ (Equation (1)).

### 3.2. Canonical States of Random Propagation

In our previous work, we described in detail how to properly model the propagation of malware by random attacks, accounting for the disparity between the number of attempted infections and actual infections (equal or smaller, due to the possibility of failed attempts). We identified two sources of inefficiency during attempted infections by random uncoordinated attacks: repetition of targets by different attacking devices and attacks performed on already-infected devices. According to these possibilities, we defined attacks as **efficient** and **wasted** and defined a maximum of four types of infection attacks, summarized below:

1. Attacking a susceptible device for the first time (efficient).
2. Making a concomitant attack on a susceptible device when the other attack failed (efficient).
3. Making a concomitant attack on a susceptible device when the other attack is successful (wasted).
4. Attacking another infected device (wasted).

For modeling purposes, this can be translated as existing only one type of attacker (infected devices), but four types of targets (one for each of the scenarios above). They are represented in Figure 1.

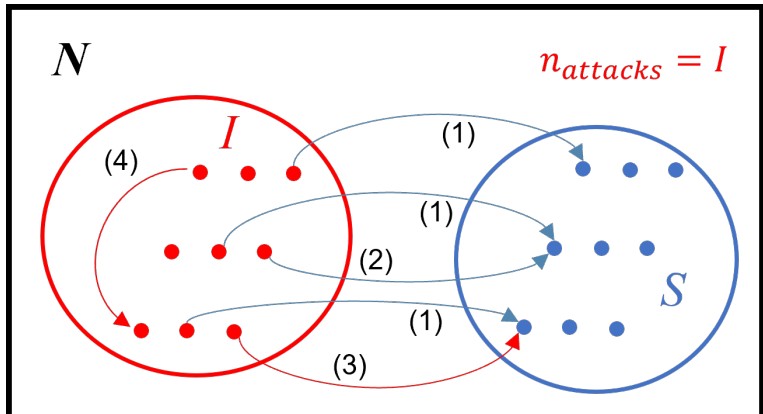

**Figure 1.** The four types of attack in a scenario of random targeting. Numbers refer to the list presenting the types of attacks. Blue arrows: efficient attacks. Red arrows: wasted attacks.

These four types of targets are equivalent to different states that a target device can display when it is chosen for an attack. However, the different number of states in various epidemic models creates a dilemma in directly representing the four scenarios of random attack. For instance, models with fewer than four states, such as Equations (2a)–(2c), cannot fully track the changes in all four target populations. On the other hand, models with more than four states, such as Equations (3a)–(3e), may raise questions about whether the four target populations described in the previous subsection can accurately represent malware propagation in such a system. But as pointed out, the modeling of malware propagation is just a sub-aspect of modeling. The standard nonlinear model is reused by very different epidemic models, but nonetheless still holds the same level of precision, so far considered satisfactory by the literature. All this means that the four scenarios of attack represent abstract states related to the phenomenon of propagation and are independent of other modeling aspects. Therefore, these states will be called hereafter the "canonical states of random propagation", since they significantly facilitate the derivation of mathematical expressions for the rate of change of random infection, regardless of the number of system states or their relationship. These canonical states are described in Table 2:

Together they represent all devices in a network, being another way of representing all populations of devices, as Equation (7) points out.

$$N = S_1[T] + S_f[T] + S_i[T] + C_a[T] \tag{7}$$

where $S_1$ are susceptible devices attacked for the first time (victim of attack Type 1); $S_f$ are susceptible devices in the process of concomitant attacks, but not infected yet (victim of attack Type 2); $S_i$ are susceptible devices in the process of concomitant attacks of which one already succeeded (victim of attack Type 3); and $C_a[T]$ are already-infected devices that are also attacking the network (victim of attack Type 4). $C_a[T]$ also represents any device whose state cannot change after a malware attack for any given reason (for example, permanently immunized devices or devices inaccessible to malware).

**Table 2.** Canonical States of Random Propagation.

| | |
|---|---|
| $S_1[T]$ | Not attacked this turn |
| $S_f[T]$ | Attacked but not infected yet |
| $S_i[T]$ | Attacked and infected this turn |
| $C_a[T]$ | All compromised/unchanging states regarding infection |

Our use of the word "canonical" is loosely borrowed from linear algebra: given a coordinate vector space, every vector inside it is described by a linear combination of a set of linearly independent vectors, called the basis of the vector space. There are infinitely possible bases for vector spaces, but one set of vectors is the simplest possible: vectors with unitary components in every perpendicular, linearly independent dimension. For example, given the $\mathbf{R}^3$ vector space, its canonical basis is the set of vectors $v_i$ below:

$$v_1 = (1, 0, 0)^T \qquad v_2 = (0, 1, 0)^T \qquad v_3 = (0, 0, 1)^T \tag{8}$$

The advantage of canonical bases is that the representation of vectors in the space becomes simple and intuitive. There is no further simplification than the usage of the canonical basis to represent vectors. Drawing an analogy to this, the canonical states of random propagation are also the simplest way to divide the total population of devices $N$ into intuitive states that facilitate the derivation of the dynamics of random propagation. Another similarity with linear algebra is the convenience of transforming one reference to the canonical form to perform easier calculations and then transforming back if the results are needed in the original reference. When calculating the exponential of matrixes, for example, it is common to diagonalize the matrix and calculate only the exponential of the diagonal elements, then transform the matrix back into the original form. In the same way, epidemic states can be transformed into canonical states of random propagation to simplify the derivation of random propagation dynamics and then are transformed back into epidemic states. As will be shown, systems with less than four states need to divide one condensed state into two or more canonical states, while systems with more than four states need to group their additional states into one or more of the four canonical states and proceed the derivation of random propagation dynamics using our methodology.

A final advantage of the canonical states is that, similarly to what is carried out with the standard nonlinear model, usually it is not necessary to repeat the entire derivation procedure, which is rather burdensome. Instead, the expression of random propagation can be modified intuitively to include more states and adapted to fit the complete system dynamics. Examples will be given in the malware epidemic models of this paper.

In the sequence, the random propagation dynamics will be derived and inserted in the complete system dynamics of three epidemic models, demonstrating the flexibility of using our model. In later sections, all models will be validated by comparing them with an equivalent simulation.

## 4. Modeling Cases

### 4.1. Derivation Case 1: SIS Model

The SIS model is the simplest cyclic compartmental model. It represents a closed cycle where infected individuals can recover and the total population can be completely cleaned of malware (see Figure 2 for the state transitions of the SIS model). Its states are only two:

$$X_{sis}[T] = (S[T], \ I[T])^T \tag{9}$$

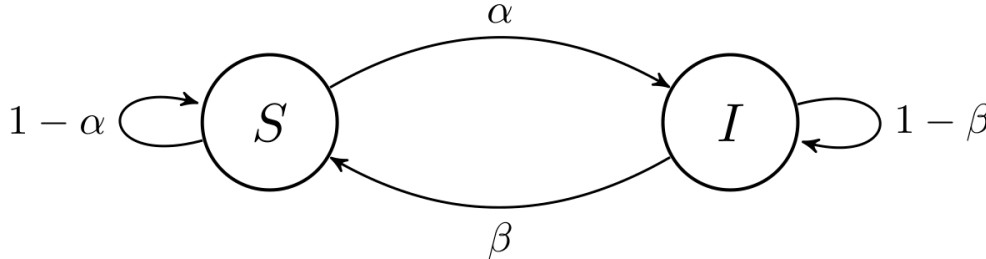

**Figure 2.** Markov process of SIS model, the simplest compartmental model.

The dynamical equation of the SIS system is then extended from Equation (1) into Equations (10a) and (10b). Note how $\Delta X$ is a composition of the infection term $I_n$ and the mitigation term $M_n$. This section will derive $I_n$ to find $X_{sis}[T+1]$.

$$\begin{cases} S[T+1] - S[T] = -I_{n/sis}[T] + M_n[T] & \text{(10a)} \\ I[T+1] - I[T] = I_{n/sis}[T] - M_n[T] & \text{(10b)} \end{cases}$$

#### 4.1.1. SIS Derivation by Methodology of Attack-by-Attack Evolution of Populations

This system has less than four states; therefore, it is necessary to divide them into four canonical states to derive the exact Markov chain of random propagation. The state $I[T]$ is directly translated into $C_a[T]$, but $S[T]$ needs to be subdivided into the other canonical states, according to Equation (11).

$$S[T] = S_1[T] + S_f[T] + S_i[T] \tag{11}$$

$$I[T] = C_a[T] \tag{12}$$

Considering that $S_i$ devices will waste any further attack aimed at them, separating them from the total number of susceptible devices reveal the real availability of $S$ for infection after every attack in a turn (which is $S_1 + S_f$). But since our real interest is the total number of new infections at the end of the turn, the calculation $S_i$ after all attacks in a turn directly yields $I_n$ for our proposed Markov chain (Equation (13)).

$$I_n[T] = {}^I S_i[T] \tag{13}$$

where ${}^I S_i[T]$ represents $S_i[T]$ after all attacks performed in a turn (one attack per infected device, in total $I$ attacks).

After all attacks of the current turn are performed, the newly infected devices $I_n[T] = S_i[T]$ will be moved from $S$ to $I$ to compose $S[T+1]$ and $I[T+1]$. If there is no mitigation, $I[T+1]$ is given by Equation (14):

$$I[T+1] - I[T] = I_n[T] \tag{14}$$

If there is mitigation, $I[T+1]$ is given by Equation (15):

$$I[T+1] - I[T] = I_n[T] - M_n[T] \tag{15}$$

Having established all possibilities of random attacks and how they are used to calculate $I_n$, the development of our exact Markov chain model can be now presented. To calculate $S_i$, it is necessary to analyze the evolution of all four populations during a single turn and repeat. In a real-world scenario, attacks can happen at the same time, but an order of attack will be assumed to facilitate the derivation of the Markov Chain. The order is the sequence of network addresses. Because the size of the populations in Equation (11) will change after every attack, another dependency on the states is necessary: $k$, standing for the $k_{th}$ attack in a turn. As an example, $^kS_1[T]$ is the size of the population of susceptible devices never attacked this turn, evaluated on turn $T$ after the $k_{th}$ attack (to clean up notation, from now on, the $T$ dependency will be omitted in some equations with the superscript $^k$, because $k$ is always considered inside the same turn). This sequential analysis of attacks will be used only to derive the Markov chain. During evaluation of the derived Markov chain between two turns, the change of states will happen at once as if all attacks were made at the same time. But note that a susceptible device that was infected this turn cannot immediately assume the infected state and display infected behavior. It can only do so in the next turn. Likewise, when the control server performs the mitigation of malware on an infected device, it will return to the susceptible state only in the next turn. The procedure of derivation is listed below.

Formulation of Target Probabilities

Firstly, it is necessary to define the probability of the $k_{th}$ attacker choosing each one of the four populations. The vector containing these four probabilities will be called "Target Probabilities". To define it, Equation (9) is expanded using Equation (11), creating the vector of canonical states $^kX_c$. The Target Probabilities of the $k_{th}$ attack will be simply the vector $X_c$ before the current attack: $^{k-1}X_c$. This probability is represented by $P(^kX_c)$ in Equation (17).

$$^kX_c = \left( \frac{^kS_1}{N}, \frac{^kS_f}{N}, \frac{^kS_i}{N}, \frac{^kC_a}{N} \right)^T \tag{16}$$

$$P(^kX_c) = {}^{k-1}X_c \tag{17}$$

Formulation of Evolution of Populations

Secondly, it is necessary to determine the new size of a population in case the $k_{th}$ attacker targets any member of the four populations. This is carried out once for every scenario of attack (four) for each population (also four). In total, there will be 16 calculations represented by $^kS_{1|j}$, $^kS_{f|j}$, $^kS_{i|j}$ and $^kC_{a|j}$, where $j$ is the index for the scenario of attack. These quantities will be called "Evolution of Populations" (see in Table 3 the form that these quantities take, both as a recursive formula and as its evaluation for different values of $k$). After a wasted attack, the Evolution of Populations will not exhibit any change (since no state was changed), but after an efficient attack, a single device will be removed from one population and given to another. In Table 3, the absence of change is shown in the two columns under "Evolution of Populations (Wasted Attacks)", while the cases with changes are shown in the two columns under "Evolution of Populations (Efficient Attacks)". In the "Efficient Attacks" cases, note how the "If $S_1$ is Attacked" case removes one device from $S_1$ and adds a fraction of 1 to both $S_f$ and $S_i$. This happens because Table 3 shows the stochastic average size of a population after an attack. The average for the Evolution of Populations is calculated by summing the two possible outcomes: new size after successful and after failed infection. That is why $^kS_{f|j}$ and $^kS_{i|j}$ shows two components: one weighted by $\alpha$ and another weighted by $1 - \alpha$. The first is the size of the population in the case of a successful attack, while the second is the size in the case of a failed attack.

**Table 3.** Evolution of SIS populations. Grey cells: iterative formula to calculate all possible state changes after every attack. Yellow cells: calculations of final expression of Population Sizes for $k = 0, 1, 2, 3$.

| | Evolution of Populations (Efficient Attacks) | | Evolution of Populations (Wasted Attacks) | | Population Size after $k_{th}$ Attack | |
|---|---|---|---|---|---|---|
| | If $S_1$ is attacked (Success or Failure) | If $S_f$ is attacked (Success or Failure) | If $S_i$ is attacked (Always Failure) | If $C_a$ is attacked (Always Failure) | Pop | Size |
| | $j = 1$ | $j = 2$ | $j = 3$ | $j = 4$ | | |
| ${}^kS_{1\mid j}$ | ${}^{k-1}S_1 - 1$ | ${}^{k-1}S_1$ | ${}^{k-1}S_1$ | ${}^{k-1}S_1$ | ${}^kS_1[T]$ | $\sum_{j=1}^{4} P\left({}^kX_c(j)\right) {}^kS_{1\mid j}$ |
| ${}^kS_{f\mid j}$ | $(1-\alpha)\left({}^{k-1}S_f + 1\right) + \alpha\,{}^{k-1}S_f$ | $(1-\alpha)\,{}^{k-1}S_f + \alpha\left({}^{k-1}S_f - 1\right)$ | ${}^{k-1}S_f$ | ${}^{k-1}S_f$ | ${}^kS_f[T]$ | $\sum_{j=1}^{4} P\left({}^kX_c(j)\right) {}^kS_{f\mid j}$ |
| ${}^kS_{i\mid j}$ | $(1-\alpha)\,{}^{k-1}S_i + \alpha\left({}^{k-1}S_i + 1\right)$ | $(1-\alpha)\,{}^{k-1}S_i + \alpha\left({}^{k-1}S_i + 1\right)$ | ${}^{k-1}S_i$ | ${}^{k-1}S_i$ | ${}^kS_i[T]$ | $\sum_{j=1}^{4} P\left({}^kX_c(j)\right) {}^kS_{i\mid j}$ |
| ${}^kC_{a\mid j}$ | ${}^{k-1}I$ | ${}^{k-1}I$ | ${}^{k-1}I$ | ${}^{k-1}I$ | ${}^kC_a[T]$ | $\sum_{j=1}^{4} P\left({}^kX_c(j)\right) {}^kC_{a\mid j}$ |
| **Target Probabilities** | | | | | | |
| | $j = 1$ | $j = 2$ | $j = 3$ | $j = 4$ | | |
| $P\left({}^kX_c(j)\right)$ | $\left({}^{k-1}S_1\right)/N$ | $\left({}^{k-1}S_f\right)/N$ | $\left({}^{k-1}S_i\right)/N$ | $I/N$ | | |
| **Before any attack ($k = 0$)** | | | | | | |
| ${}^0S_{1\mid j}$ | - | - | - | - | ${}^0S_1$ | $S$ |
| ${}^0S_{f\mid j}$ | - | - | - | - | ${}^0S_f$ | $0$ |
| ${}^0S_{i\mid j}$ | - | - | - | - | ${}^0S_i$ | $0$ |
| ${}^0C_{a\mid j}$ | - | - | - | - | ${}^0C_a$ | $I$ |
| $P\left({}^0X_c(j)\right)$ | - | - | - | - | | |
| **After the 1st attack ($k = 1$)** | | | | | | |
| ${}^1S_{1\mid j}$ | $S - 1$ | - | - | $S$ | ${}^1S_1$ | $S - \dfrac{S}{N}$ |
| ${}^1S_{f\mid j}$ | $1 - \alpha$ | - | - | $0$ | ${}^1S_f$ | $(1-\alpha)\dfrac{S}{N}$ |
| ${}^1S_{i\mid j}$ | $\alpha$ | - | - | $0$ | ${}^1S_i$ | $\alpha\dfrac{S}{N}$ |
| ${}^1C_{a\mid j}$ | $I$ | $I$ | $I$ | $I$ | ${}^1C_a$ | $I$ |
| $P\left({}^1X_c(j)\right)$ | $S/N$ | $0$ | $0$ | $I/N$ | | |

**Table 3.** *Cont.*

| | Evolution of Populations (Efficient Attacks) | | Evolution of Populations (Wasted Attacks) | | Population Size after k$_{th}$ Attack | |
|---|---|---|---|---|---|---|
| | If $S_1$ is attacked (Success or Failure) | If $S_f$ is attacked (Success or Failure) | If $S_i$ is attacked (Always Failure) | If $C_a$ is attacked (Always Failure) | Pop | Size |
| | $j = 1$ | $j = 2$ | $j = 3$ | $j = 4$ | | |
| **After the 2nd attack ($k = 2$)** | | | | | | |
| $^2S_{1|j}$ | $S - \dfrac{S}{N} - 1$ | $S - \dfrac{S}{N}$ | $S - \dfrac{S}{N}$ | $S - \dfrac{S}{N}$ | $^2S_1$ | $S - 2\dfrac{S}{N} + \dfrac{S}{N^2}$ |
| $^2S_{f|j}$ | $(1-\alpha)\left(\dfrac{S}{N} - \alpha\dfrac{S}{N} + 1\right) + \alpha\left(\dfrac{S}{N} - \alpha\dfrac{S}{N}\right)$ | $(1-\alpha)\left(\dfrac{S}{N} - \alpha\dfrac{S}{N}\right) + \alpha\left(\dfrac{S}{N} - \alpha\dfrac{S}{N} - 1\right)$ | $(1-\alpha)\dfrac{S}{N}$ | $(1-\alpha)\dfrac{S}{N}$ | $^2S_f$ | $2(1-\alpha)\dfrac{S}{N} - (1-\alpha^2)\dfrac{S}{N^2}$ |
| $^2S_{i|j}$ | $(1-\alpha)\left(\alpha\dfrac{S}{N}\right) + \alpha\left(\alpha\dfrac{S}{N} + 1\right)$ | $(1-\alpha)\left(\alpha\dfrac{S}{N}\right) + \alpha\left(\alpha\dfrac{S}{N} + 1\right)$ | $\alpha\dfrac{S}{N}$ | $\alpha\dfrac{S}{N}$ | $^2S_i$ | $2\alpha\dfrac{S}{N} + 2\alpha^2\dfrac{S}{N^2}$ |
| $^2C_{a|j}$ | $I$ | $I$ | $I$ | $I$ | $^2C_a$ | $I$ |
| $P\left(^2X_c(j)\right)$ | $\left(S - \dfrac{S}{N}\right)/N$ | $\left((1-\alpha)\dfrac{S}{N}\right)/N$ | $\left(\alpha\dfrac{S}{N}\right)/N$ | $I/N$ | | |
| **After the 3rd attack ($k = 3$)** | | | | | | |
| $^3S_{1|j}$ | $S - 2\dfrac{S}{N} + \dfrac{S}{N^2} - 1$ | $S - 2\dfrac{S}{N} + \dfrac{S}{N^2}$ | $S - 2\dfrac{S}{N} + \dfrac{S}{N^2}$ | $S - 2\dfrac{S}{N} + \dfrac{S}{N^2}$ | $^3S_1$ | $S - 3\dfrac{S}{N} + 3\dfrac{S}{N^2} - \dfrac{S}{N^3}$ |
| $^3S_{f|j}$ | $(1-\alpha)\left(2(1-\alpha)\dfrac{S}{N} - (1-\alpha^2)\dfrac{S}{N^2} + 1\right)$ $+\alpha\left(2(1-\alpha)\dfrac{S}{N} - (1-\alpha^2)\dfrac{S}{N^2}\right)$ | $(1-\alpha)\left(2(1-\alpha)\dfrac{S}{N} - (1-\alpha^2)\dfrac{S}{N^2}\right)$ $+\alpha\left(2(1-\alpha)\dfrac{S}{N} - (1-\alpha^2)\dfrac{S}{N^2} + 1\right)$ | $2(1-\alpha)\dfrac{S}{N} - (1-\alpha^2)\dfrac{S}{N^2}$ | $2(1-\alpha)\dfrac{S}{N} - (1-\alpha^2)\dfrac{S}{N^2}$ | $^3S_f$ | $3(1-\alpha)\dfrac{S}{N} - 3(1-\alpha^2)\dfrac{S}{N^2} + (1-\alpha^3)\dfrac{S}{N^3}$ |
| $^3S_{i|j}$ | $(1-\alpha)\left(2\alpha\dfrac{S}{N} + 2\alpha^2\dfrac{S}{N^2}\right)$ $+\alpha\left(2\alpha\dfrac{S}{N} + 2\alpha^2\dfrac{S}{N^2} - 1\right)$ | $(1-\alpha)\left(2\alpha\dfrac{S}{N} + 2\alpha^2\dfrac{S}{N^2}\right)$ $+\alpha\left(2\alpha\dfrac{S}{N} + 2\alpha^2\dfrac{S}{N^2} + 1\right)$ | $2\alpha\dfrac{S}{N} + 2\alpha^2\dfrac{S}{N^2}$ | $2\alpha\dfrac{S}{N} + 2\alpha^2\dfrac{S}{N^2}$ | $^3S_i$ | $3\alpha\dfrac{S}{N} - 3\alpha^2\dfrac{S}{N^2} + \alpha^3\dfrac{S}{N^3}$ |
| $^3C_{a|j}$ | $I$ | $I$ | $I$ | $I$ | $^3C_a$ | $I$ |
| $P\left(^3X_c(j)\right)$ | $\left(S - 2\dfrac{S}{N} + \dfrac{S}{N^2}\right)/N$ | $\left(2(1-\alpha)\dfrac{S}{N} - (1-\alpha^2)\dfrac{S}{N^2}\right)/N$ | $\left(2\alpha\dfrac{S}{N} + 2\alpha^2\dfrac{S}{N^2}\right)/N$ | $I/N$ | | |

Formulation of New Size of Populations after an Attack

Finally, the probabilistic size of each population is calculated after the $k_{th}$ attack using Equations (18)–(21). They are the internal product of Target Probabilities and Evolution of Populations. This internal product is calculated once for each population. Note that, given any of the four populations, its Target Probability is the same in all scenarios of attack (because of Equation (17)), but its Evolution of Population is different for every scenario of attack (because of $j$ and Table 3). The Population Size of all four canonical states after the $k_{th}$ attack is shown on the rightmost side of Table 3 in the yellow cells after the Evolution of Populations are weighted by the Target Probabilities and are all summed into the final Population Size.

$$^kS_1[T] = \sum_{j=1}^{4} P(^kX_c(j))^kS_{1|j} \tag{18}$$

$$^kS_f[T] = \sum_{j=1}^{4} P(^kX_c(j))^kS_{f|j} \tag{19}$$

$$^kS_i[T] = \sum_{j=1}^{4} P(^kX_c(j))^kS_{i|j} \tag{20}$$

$$^kC_a[T] = \sum_{j=1}^{4} P(^kX_c(j))^kC_{a|j} \tag{21}$$

Expanding these series for $k > 3$ (more than three attacks) becomes burdensome and yields increasingly big expressions. To deduce a simplified expression for any $k$, we expanded Equations (18)–(21) for each $k$ from 0 to 3 and looked for mathematical patterns. The calculations are shown in Table 3. The table is divided into five vertical sections: the top section shows the recursive formulas with respect to $k$, while the second to bottom sections show the resulting expressions from $k = 0$ to $k = 3$. Each section is further divided into three subtables: top-left is the subtable containing the expression for the Evolution of Populations; below it is the subtable containing the Target Probabilities; and the top-right is the subtable containing the Population Size After $k_{th}$ Attack. Table 3 takes us into every step necessary to calculate the desired Population Size After $k_{th}$ Attack. As Equations (18)–(21) indicate, we multiply the Target Probabilities by Evolutions of Populations ($j$ by $j$, from 1 to 4), and sum the four multiplications to find its Population Size. Looking at the Population Size of all canonical states for $k = 0, 1, 2, 3$ (last column, marked in yellow), it can be seen that after every attack, a new term was added to the Population Size. Checking the coefficients of $^0S_1$, $^1S_1$, $^2S_1$, and $^3S_1$, we find they are, respectively, the same as the elements of Rows 1, 2, 3, and 4 of the extended Pascal triangle (which has the same elements as Pascal's triangle but with consecutive elements alternating between positive and negative values). Figure 3 shows the first four rows of Pascal's triangle, while Figure 4 shows the same for the extended Pascal triangle

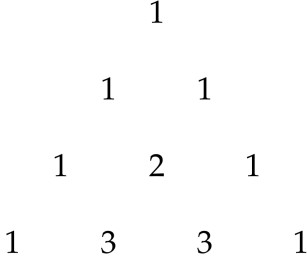

**Figure 3.** Pascal's triangle.

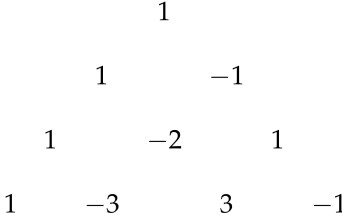

**Figure 4.** Extended Pascal triangle.

The elements of Pascal's triangle can be described by the operator $\binom{n}{p}$, which represents the number of ways to choose $p$ elements from a set of $n$ distinct elements. Multiplying it by $(-1)^i$ will modify its sign alternately along the rows, describing the elements of the extended Pascal triangle. With this operator, we formulate a simplified expression for the Population Size of the canonical states after an arbitrary number of $I$ attacks, shown in Equations (22)–(25):

$$^{I}S_1[T] = N \sum_{i=0}^{I} (-1)^i \binom{I}{i} \frac{S}{N^{i+1}} \tag{22}$$

$$^{I}S_f[T] = N \sum_{i=1}^{I} (-1)^i \binom{I}{i} I(1-\alpha^i) \frac{S}{N^{i+1}} \tag{23}$$

$$^{I}S_i[T] = N \sum_{i=1}^{I} (-1)^i \binom{I}{i} I\alpha^i \frac{S}{N^{i+1}} \tag{24}$$

$$^{I}C_a[T] = C_a[T] \tag{25}$$

Note how only $S_1$ has a summation that starts with $i = 0$, while $S_f$ and $S_i$ start with $i = 1$. This highlights how $S_f$ and $S_i$ are subtractions of $S_1$ that grow in time as more attacks are performed. But despite the exactness of the derivation above, the model is still computationally heavy. By using the Binomial Theorem it is possible to further simplify the equations. The coefficients of the $k$ expanded summation of $S_1$ are equal to the elements of the $k+1$ row of the extended Pascal triangle (the version with alternating negative elements). Since the theorem posits that every row can be simplified to the same binomial and that the exponent is equivalent to the number of the row, Equations (22)–(25) can be transformed into the following ones:

$$^{I}S_1[T] = S[T] \left( 1 - \frac{1}{N} \right)^{I} \tag{26}$$

$$^{I}S_f[T] = S[T] \left( \left( 1 - \frac{\alpha}{N} \right)^{I} - \left( 1 - \frac{1}{N} \right)^{I} \right) \tag{27}$$

$$^{I}S_i[T] = S[T] - S[T] \left( 1 - \frac{\alpha}{N} \right)^{I} \tag{28}$$

$$^{I}C_a[T] = C_a[T] \tag{29}$$

Substituting Equation (28) into Equation (13), the exact Markov chain is found for random malware propagation:

$$I_{n/sis}[T] = S[T] \left( 1 - \left( 1 - \frac{\alpha}{N} \right)^{I[T]} \right) \tag{30}$$

Compare it with the SIS standard nonlinear model (Equation (5)). The three original mathematical operations become five, with the benefit of an exact calculation that eliminates errors as big as $\sim$10% of a population, as we will see in Section 6.2.

### 4.2. Derivation Case 2: SEIRS Model with Delay

The compartmental model SEIRS is a popular template used in many studies with specific modifications to the modeled network [25,26,31]. It presents a delay mechanism in both infection and mitigation. After being infected, devices are first brought to a state of temporary incubation (called exposed and represented by *E*), where they remain asymptomatic for some turns and do not replicate the infection. After being cleaned, devices are brought to a state of temporary immunity (called recovered and represented by *R*) and cannot be infected again for some turns. In most models, devices stay in *E* and *R* for a fixed number of turns and then finish transitioning to the final state. *E* takes $\omega$ turns to transition to *I*, while *R* takes $\tau$ turns to transition to *S*. Its states are four:

$$X_{seirs}[T] = (S[T],\ E[T],\ I[T],\ R[T])^T \tag{31}$$

See Figure 5 for the state machine of SEIRS. An important observation is that the complete model is not a Markov chain because it depends on older states than the current one (the transition of *E* to *I* and *R* to *S* are mediated by delays). But the propagation model $I_{n/seirs}$ that we are modeling depends only on the previous state; therefore, it is a Markov chain by itself.

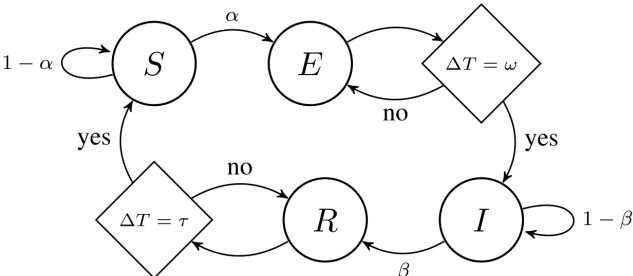

**Figure 5.** State machine of SEIRS model.

The dynamical equation of the SEIRS system is developed from Equation (1) into Equations (32a)–(32d).

$$
\begin{cases}
S[T+1] - S[T] = -I_{n/seirs}[T] + M_n[T - \tau] & \text{(32a)} \\[2mm]
E[T+1] - E[T] = I_{n/seirs}[T] - I_{n/seirs}[T - \omega] & \text{(32b)} \\[2mm]
I[T+1] - I[T] = I_{n/seirs}[T - \omega] - M_n[T] & \text{(32c)} \\[2mm]
R[T+1] - R[T] = M_n[T] - M_n[T - \tau] & \text{(32d)}
\end{cases}
$$

Although this system has four states, to derive the exact Markov chain of random propagation, it is necessary to consider them only in terms of efficient and wasted attacks. From the four states, only *S* can possibly change states after an attack. Devices from *E*, *I*, and *R* change states due to other dynamics, not infection attempts (*I* is due to mitigation, and *E* or *R* are due to the time delay after infection or mitigation). Therefore, *S* will be divided into the canonical states $S_1$, $S_f$, and $S_i$ (Equation (33))

$$S[T] = S_1[T] + S_f[T] + S_i[T] \tag{33}$$

and *E*, *I*, and *R* will be grouped into $C_a[T]$ (Equation (34)).

$$C_a[T] = E[T] + I[T] + R[T] \tag{34}$$

Moreover, the number of attacks will be determined by $I$ only, since $E$ devices are already compromised but are not yet propagating the infection.

4.2.1. Methodology 1 of Derivation of SEIRS Propagation: Modification of the Basic Propagation Term $I_{n/sis}$

There are two important factors to take into account when modifying the basic propagation term $I_{n/sis}$:

1.  Which epidemic states are equivalent to $S_1$, $S_f$ and $S_i$.
2.  How to parameterize the number of malware attacks in terms of epidemic states.

The first factor is due to the fact that from the four canonical states, only $S_1$, $S_f$, and $S_i$ actually change during attacks. The equivalence between the epidemic states, and these three canonical states will affect $S$ in the basic formula of $I_{n/sis}$ (Equation (30)). The second factor is simply related to the exponent $I$ in the formula of $I_{n/sis}$: does only $I$ contribute to the number of attacks, or does any other epidemic state also contribute? If it is the latter case, this needs to be reflected in the binomial exponent of $I_n$.

Moving on to the modification of $I_{n/sis}$ for the SEIRS model, at first, big changes could be expected since the SEIRS delays introduce a complex dynamic in the interaction between $S$ and $I$. They also affect malware propagation significantly by reducing the populations available to infection. But when the two factors above are considered, the SEIRS model shows characteristics equivalent to the SIS model: only $S$ is transformed into $S_1$, $S_f$, and $S_i$, and only $I$ is considered in the number of malware attacks. Therefore, the formula for $I_{n/seirs}$ will be exactly the same as Equation (30):

$$I_{n/seirs}[T] = S[T]\left(1 - \left(1 - \frac{\alpha}{N}\right)^{I[T]}\right) \tag{35}$$

The delayed transition from $E$ to $I$ is modeled by the simple subtraction of the delay $\omega$ from $T$ in $I_{n/seirs}$, leading to Equation (36):

$$I_{n/seirs}[T - \omega] = S[T - \omega]\left(1 - \left(1 - \frac{\alpha}{N}\right)^{I[T-\omega]}\right) \tag{36}$$

Compare them with the SEIRS standard nonlinear model of propagation, Equations (37) and (38):

$$I_{n/seirs/std}[T] = \alpha S[T]\frac{I[T]}{N} \tag{37}$$

$$I_{n/seirs/std}[T - \omega] = \alpha S[T - \omega]\frac{I[T - \omega]}{N} \tag{38}$$

The normal SEIRS mitigation (non-delayed) is the same as the SIS one (Equation (6)). The delayed transition from $R$ to $I$ is modeled by subtracting $\tau$ from $T$ in the equation for $M_n$.

$$M_n[T - \tau] = \beta I[T - \tau] \tag{39}$$

4.2.2. Methodology 2 of Derivation of SEIRS Propagation: Formulation of Evolution of Populations

If one wants to derive the propagation term $I_n$ more precisely than by intuitive modification of $I_{n/sis}$, then the same recursive formulas shown in Table 3 can be applied in the SEIRS model. To do that, Equations (33) and (34) can be analyzed to see how the four epidemic states $S$, $E$, $I$, and $R$ will be transformed into the four canonical states of propagation, then apply the same formulas of adding or removing one device from the populations of canonical states as explained in "Formulation of Evolution of Populations" (Section 4.1.1). For the SEIRS model, only the recursive formula will be shown, but the evaluation will be omitted for the sake of conciseness.

As Table 4 shows, the only difference in the iterative formulas between SIS and SEIRS models is the unchanging canonical state $C_a$, which does not affect the derivation of $I_n$. When the mathematical patterns of the first three attacks are extrapolated to all attacks, the number of attacks in turn continues to be $I[T]$ as in SIS. Therefore, the results of Table 4 are exactly the same as the SIS table, and the terms $I_{n/sis}$ and $I_{n/seirs}$ are exactly the same.

**Table 4.** Evolution of SEIRS populations. Grey cells: iterative formula to calculate all possible state changes after every attack.

| | Evolution of Populations (Efficient Attacks) | | Evolution of Populations (Wasted Attacks) | | Population Size After $k_{th}$ Attack | |
|---|---|---|---|---|---|---|
| | If $S_1$ is attacked (Success or Failure) | If $S_f$ is attacked (Success or Failure) | If $S_i$ is attacked (Always Failure) | If $C_a$ is attacked (Always Failure) | Pop | Size |
| | $j=1$ | $j=2$ | $j=3$ | $j=4$ | | |
| $^kS_{1|j}$ | $^{k-1}S_1 - 1$ | $^{k-1}S_1$ | $^{k-1}S_1$ | $^{k-1}S_1$ | $^kS_1[T]$ | $\sum_{j=1}^{4} P(^kX_c(j))\,^kS_{1|j}$ |
| $^kS_{f|j}$ | $(1-\alpha)(^{k-1}S_f + 1) + \alpha\,^{k-1}S_f$ | $(1-\alpha)^{k-1}S_f + \alpha(^{k-1}S_f - 1)$ | $^{k-1}S_f$ | $^{k-1}S_f$ | $^kS_f[T]$ | $\sum_{j=1}^{4} P(^kX_c(j))\,^kS_{f|j}$ |
| $^kS_{i|j}$ | $(1-\alpha)^{k-1}S_i + \alpha(^{k-1}S_i + 1)$ | $(1-\alpha)^{k-1}S_i + \alpha(^{k-1}S_i + 1)$ | $^{k-1}S_i$ | $^{k-1}S_i$ | $^kS_i[T]$ | $\sum_{j=1}^{4} P(^kX_c(j))\,^kS_{i|j}$ |
| $^kC_{a|j}$ | $^{k-1}E + ^{k-1}I + ^{k-1}R$ | $^{k-1}E + ^{k-1}I + ^{k-1}R$ | $^{k-1}E + ^{k-1}I + ^{k-1}R$ | $^{k-1}E + ^{k-1}I + ^{k-1}R$ | $^kC_a[T]$ | $\sum_{j=1}^{4} P(^kX_c(j))\,^kC_{a|j}$ |
| **Target Probabilities** | | | | | | |
| | $j=1$ | $j=2$ | $j=3$ | $j=4$ | | |
| $P(^kX_c(j))$ | $(^{k-1}S_1)/N$ | $(^{k-1}S_f)/N$ | $(^{k-1}S_i)/N$ | $(E + I + R)/N$ | | |

### 4.2.3. Note on the Evaluation of Complete SEIRS Dynamics

Although the propagation model $I_n$ is the same for SIS and SEIRS, there is an important difference in the complete system dynamics (Equations (32a)–(32d)): $I_{n/seirs}$ is a Markov chain, but the system dynamics is not. The delays make the system dependent on older states than the current one. When evaluating the system dynamics, it will be necessary to store previous states and to index the proper turn of the delayed terms with $[T - \omega]$ or $[T - \tau]$. Apart from this, SIS and SEIRS models are basically different systems using the same terms $I_n$ and $M_n$ in different ways (like the standard nonlinear model of random propagation).

### 4.3. Derivation Case 3: 2SEIS Model of Heterogeneous Network

The 2SEIS model is proposed in this work with the didactic purpose of demonstrating how to derive the random propagation model for a heterogeneous network. In this network, there are two types of devices: CCTV devices (indicated by $S$) and mobile devices (indicated by $S'$). The former behaves like the SIS model and transitions from $S$ to $I$ immediately after infection, while the latter has the "incubation" delay of the SEIRS model and transitions from $S'$ to $E'$ first, then to $I'$. However, this different behavior is caused by a single malware that can infect both devices and be passed from one type to the other. Another motivation for proposing this model is to explore characteristics of heterogeneous dynamics in modeling, aiming to better relate epidemic models with the increasing complexity of cross-platform Internet connectivity that is seen today.

Like in the SEIRS model, the delay from exposed to infected is represented by $\omega$ turns of delay, and therefore, the complete model is not a Markov chain while the propagation model $I_{n/2seis}$ is. See Figure 6 for the state machine of 2SEIS. There are five states in total:

$$X_{2seis}[T] = \big(S[T],\ I[T],\ S'[T],\ E'[T],\ I'[T]\big)^T \tag{40}$$

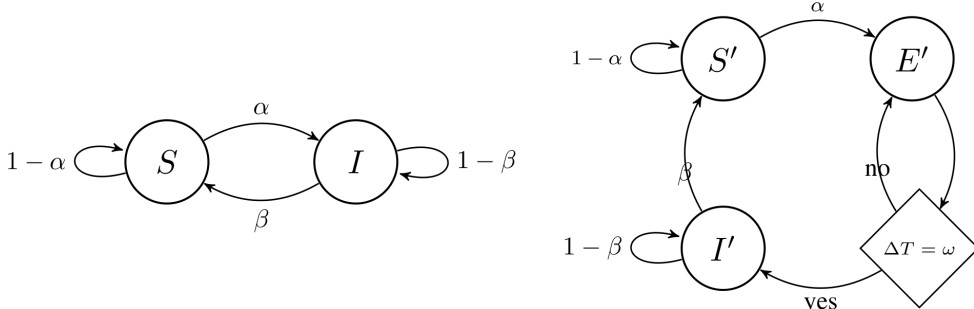

**Figure 6.** State machine of 2SEIS model.

The dynamical equation of the 2SEIS system is then extended from Equation (1) to Equations (41a)–(41e). Note how each device has its own infection and mitigation terms.

$$\begin{cases} S[T+1] - S[T] = -I_{n/2seis}[T] + M_n[T] & \text{(41a)} \\[2mm] I[T+1] - I[T] = I_{n/2seis}[T] - M_n[T] & \text{(41b)} \\[2mm] S'[T+1] - S'[T] = -I'_{n/2seis}[T] + M'_n[T] & \text{(41c)} \\[2mm] E'[T+1] - E'[T] = I'_{n/2seis}[T] - I'_{n/2seis}[T-\omega] & \text{(41d)} \\[2mm] I'[T+1] - I'[T] = I'_{n/2seis}[T-\omega] - M'_n[T] & \text{(41e)} \end{cases}$$

4.3.1. Methodology 1 of Derivation of 2SEIS Propagation: Modification of the Basic Propagation Term $I_{n/sis}$

Although this system has five states, two of them are susceptible states, one for each type of device; two are infected states, one for each type; and one is the exposed state for the mobile type. Regarding the first factor listed in Section 4.2.1, the susceptible state of each device will be divided into its own trio of susceptible canonical states, because CCTV and mobile devices behave differently with respect to infection. Regarding the second factor, consider how the same malware infects both types of devices and each type can infect the other. This translates into a synergy between $I$ and $I'$ that is reflected in the binomial exponent of the propagation terms ($I_n[T]$, $I'_n[T]$, and $I'_n[T-\omega]$). Both $I$ and $I'$ will contribute to the quantity of attacks in each device, leading to $(1-\alpha/N)^{(I+I')}$.

Paying attention to these two factors, the following propagation terms are found. Equation (42) represents random malware propagation in CCTV devices, Equation (43) represents the same in mobile devices and Equation (44) represents the delayed change of exposed mobile devices into infected ones.

$$I_{n/2seis}[T] = S[T]\left(1 - \left(1 - \frac{\alpha}{N}\right)^{I[T]+I'[T]}\right) \tag{42}$$

$$I'_{n/2seis}[T] = S'[T]\left(1 - \left(1 - \frac{\alpha}{N}\right)^{I[T]+I'[T]}\right) \tag{43}$$

$$I'_{n/2seis}[T-\omega] = S'[T-\omega]\left(1 - \left(1 - \frac{\alpha}{N}\right)^{I[T-\omega]+I'[T-\omega]}\right) \tag{44}$$

Compare them with the 2SEIS standard nonlinear model of propagation, Equations (45)–(47):

$$I_{n/2seis/std}[T] = \alpha S[T]\frac{I[T] + I'[T]}{N} \tag{45}$$

$$I'_{n/2seis/std}[T] = \alpha S'[T]\frac{I[T] + I'[T]}{N} \tag{46}$$

$$I'_{n/2seis/std}[T - \omega] = \alpha S'[T - \omega]\frac{I[T - \omega] + I'[T - \omega]}{N} \tag{47}$$

With this reasoning, the terms are readily adapted from the basic random propagation term of SIS model (Equation (30)) an the delayed term of exposed state of SEIRS model (Equation (36)).

4.3.2. Methodology 2 of Derivation of 2SEIS Propagation: Formulation of Evolution of Populations

The modification of $I_n$ for 2SEIS was minimal, something recurrent in the derivation of propagation terms for different epidemic models. But in the case of difficulty in the modification, a sure derivation of $I_{n/2seis}$ can be performed through the recursive formulas of Table 5.

**Table 5.** Evolution of 2SEIS populations. Grey cells: iterative formula to calculate all possible state changes after every attack.

| | Evolution of Populations (Efficient Attacks) | | Evolution of Populations (Wasted Attacks) | | Population Size After $k_{th}$ Attack | |
|---|---|---|---|---|---|---|
| | If $S_1$ is attacked (Success or Failure) | If $S_f$ is attacked (Success or Failure) | If $S_i$ is attacked (Always Failure) | If $C_a$ is attacked (Always Failure) | Pop | Size |
| | $j = 1$ | $j = 2$ | $j = 3$ | $j = 4$ | | |
| ${}^k S_{1\mid j}$ | ${}^{k-1}S_1 - 1$ | ${}^{k-1}S_1$ | ${}^{k-1}S_1$ | ${}^{k-1}S_1$ | ${}^k S_1[T]$ | $\sum_{j=1}^{7} P({}^k X_c(j)){}^k S_{1\mid j}$ |
| ${}^k S_{f\mid j}$ | $(1-\alpha)({}^{k-1}S_f + 1) + \alpha^{k-1}S_f$ | $(1-\alpha)^{k-1}S_f + \alpha({}^{k-1}S_f - 1)$ | ${}^{k-1}S_f$ | ${}^{k-1}S_f$ | ${}^k S_f[T]$ | $\sum_{j=1}^{7} P({}^k X_c(j)){}^k S_{f\mid j}$ |
| ${}^k S_{i\mid j}$ | $(1-\alpha)^{k-1}S_i + \alpha({}^{k-1}S_i + 1)$ | $(1-\alpha)^{k-1}S_i + \alpha({}^{k-1}S_i + 1)$ | ${}^{k-1}S_i$ | ${}^{k-1}S_i$ | ${}^k S_i[T]$ | $\sum_{j=1}^{7} P({}^k X_c(j)){}^k S_{i\mid j}$ |
| ${}^k C_{a\mid j}$ | ${}^{k-1}I + {}^{k-1}I' + {}^{k-1}E$ | ${}^{k-1}I + {}^{k-1}I' + {}^{k-1}E$ | ${}^{k-1}I + {}^{k-1}I' + {}^{k-1}E$ | ${}^{k-1}I + {}^{k-1}I' + {}^{k-1}E$ | ${}^{k-1}C_a[T]$ | $\sum_{j=1}^{7} P({}^k X_c(j)){}^k C_{a\mid j}$ |
| | If $S'_1$ is attacked (Success or Failure) | If $S'_f$ is attacked (Success or Failure) | If $S'_i$ is attacked (Always Failure) | | Pop | Size |
| | $j = 5$ | $j = 6$ | $j = 7$ | | | |
| ${}^k S'_{1\mid j}$ | ${}^{k-1}S'_1 - 1$ | ${}^{k-1}S'_1$ | ${}^{k-1}S'_1$ | | ${}^k S'_1[T]$ | $\sum_{j=1}^{7} P({}^k X_c(j)){}^k S'_{1\mid j}$ |
| ${}^k S'_{f\mid j}$ | $(1-\alpha)({}^{k-1}S'_f + 1) + \alpha^{k-1}S'_f$ | $(1-\alpha)^{k-1}S'_f + \alpha({}^{k-1}S'_f - 1)$ | ${}^{k-1}S'_f$ | | ${}^k S'_f[T]$ | $\sum_{j=1}^{7} P({}^k X_c(j)){}^k S'_{f\mid j}$ |
| ${}^k S'_{i\mid j}$ | $(1-\alpha)^{k-1}S'_i + \alpha({}^{k-1}S'_i + 1)$ | $(1-\alpha)^{k-1}S'_i + \alpha({}^{k-1}S'_i + 1)$ | ${}^{k-1}S'_i$ | | ${}^k S'_i[T]$ | $\sum_{j=1}^{7} P({}^k X_c(j)){}^k S'_{i\mid j}$ |
| **Target Probabilities** | | | | | | |
| | $j = 1$ | $j = 2$ | $j = 3$ | $j = 4$ | | |
| $P({}^k X_c(j))$ | $({}^{k-1}S_1)/N$ | $({}^{k-1}S_f)/N$ | $({}^{k-1}S_i)/N$ | $(I + I' + E)/N$ | | |
| | $j = 5$ | $j = 6$ | $j = 7$ | | | |
| $P({}^k X_c(j))$ | $({}^{k-1}S'_1)/N$ | $({}^{k-1}S'_f)/N$ | $({}^{k-1}S'_i)/N$ | | | |

Firstly, all nonsusceptible states are separated from the two susceptible states and grouped together into $C_a[T]$ because none of them can change state due to a malware attack:

$$C_a[T] = E'[T] + I[T] + I'[T] \tag{48}$$

Then, one subdivision of $S$ into the three other canonical states is created for each device. In other words:

$$S[T] = S_1[T] + S_f[T] + S_i[T] \tag{49}$$

$$S'[T] = S'_1[T] + S'_f[T] + S'_i[T] \tag{50}$$

Then, the recurrent formulas shown in Table 5 are developed, iterating for the three first attacks to see how the patterns emerge. Since there are five states, the recursive calculation is rather burdensome, but many terms cancel themselves, and the general format of the propagation term is identical for both $S$ and $S'$. It is almost the same result of $I_{n/sis}$, but when the iterations are extrapolated to the total numbers of attacks, the difference in the binomial exponent takes the form of $I + I'$, because in a single turn, there are $I + I'$ malware attacks. This leads to Equations (42)–(44) as derived in the last subsection by modification of $I_{n/sis}$.

## 5. Exact Markov Chains

### 5.1. Dtmc of Example 1: SIS Model

Substituting Equations (30) and (6) into Equations (10a) and (10b) (for the propagation term $I_n$ and the mitigation term $M_n$ respectively) yields the complete dynamics of the SIS model in Equations (51a) and (51b).

$$
\begin{cases}
S[T+1] - S[T] = -S[T]\left(1 - \left(1 - \dfrac{\alpha}{N}\right)^{I[T]}\right) + \beta I[T] & \text{(51a)} \\[4mm]
I[T+1] - I[T] = S[T]\left(1 - \left(1 - \dfrac{\alpha}{N}\right)^{I[T]}\right) - \beta I[T] & \text{(51b)}
\end{cases}
$$

### 5.2. Dtmc of Example 2: SEIRS Model

Substituting Equations (35) and (6) into Equations (32a)–(32d) for the propagation term $I_n[T]$, the mitigation term $M_n[T]$ and their respective versions with delays $I_n[T - \omega]$ and $M_n[T - \tau]$, yields the complete dynamics of the SEIRS model in Equations (52a)–(52d).

$$
\begin{cases}
S[T+1] - S[T] = -S[T]\left(1 - \left(1 - \dfrac{\alpha}{N}\right)^{I[T]}\right) + \beta I[T - \tau] & \text{(52a)} \\[4mm]
E[T+1] - E[T] = S[T]\left(1 - \left(1 - \dfrac{\alpha}{N}\right)^{I[T]}\right) - S[T - \omega]\left(1 - \left(1 - \dfrac{\alpha}{N}\right)^{I[T-\omega]}\right) & \text{(52b)} \\[4mm]
I[T+1] - I[T] = S[T - \omega]\left(1 - \left(1 - \dfrac{\alpha}{N}\right)^{I[T-\omega]}\right) - \beta I[T] & \text{(52c)} \\[4mm]
R[T+1] - R[T] = \beta I[T] - \beta I[T - \tau] & \text{(52d)}
\end{cases}
$$

### 5.3. Dtmc of Example 3: 2SEIS Model

Substituting Equations (42)–(44) and (6) into Equations (41a)–(41e) yields the complete dynamics of the 2SEIS model in Equations (53a)–(53e).

$$
\begin{cases}
S[T+1] - S[T] = -S[T]\left(1 - \left(1 - \dfrac{\alpha}{N}\right)^{I[T]+I'[T]}\right) + \beta I[T] & \text{(53a)} \\[4mm]
I[T+1] - I[T] = S[T]\left(1 - \left(1 - \dfrac{\alpha}{N}\right)^{I[T]+I'[T]}\right) - \beta I[T] & \text{(53b)} \\[4mm]
S'[T+1] - S'[T] = -S'[T]\left(1 - \left(1 - \dfrac{\alpha}{N}\right)^{I[T]+I'[T]}\right) + \beta I'[T] & \text{(53c)} \\[4mm]
E'[T+1] - E'[T] = S'[T]\left(1 - \left(1 - \dfrac{\alpha}{N}\right)^{I[T]+I'[T]}\right) - S'[T - \omega]\left(1 - \left(1 - \dfrac{\alpha}{N}\right)^{I[T-\omega]+I'[T-\omega]}\right) & \text{(53d)} \\[4mm]
I'[T+1] - I'[T] = S'[T - \omega]\left(1 - \left(1 - \dfrac{\alpha}{N}\right)^{I[T-\omega]+I'[T-\omega]}\right) - \beta I'[T] & \text{(53e)}
\end{cases}
$$

## 6. Validation Of Proposed Model

In order to demonstrate the validity of our model and compare its performance with the main model of literature, we developed a stochastic simulation that performs infections

and mitigation actions device by device. Setting the same parameters for the Markov chain models and the simulation, the results of the Markov chain and the simulation were generated independently and compared. The simulation results are independent of the Markov chain because Equations (51a)–(53e) represent the epidemic states of entire populations of devices without keeping track of the state of individual devices. The simulation, however, stores the state of every device. Moreover, the infection and detection rates behave as deterministic parameters in the equation, but in the simulation, they are real random tests performed at every attempt of infection and cleaning. By running the simulation against a sufficiently big population, the stochastic variability of the simulation is minimized, and a fair, independent comparison becomes possible. The simulation code is available on a public GitHub repository (link at the end of the paper).

Figures 7–9 show the results of the simulation of SIS, SEIRS, and 2SEIS. Although simulations were performed for many different parameters (see Table 6), only the figures of the simulation for the high rate of infection will be shown since this is the scenario with the biggest degradation of performance for the standard nonlinear model. This and other results will be discussed later in this section.

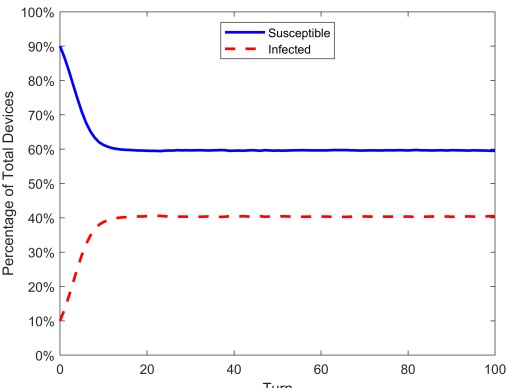

**Figure 7.** Results of simulation of models SIS for $\alpha = 90/\beta = 45$, with the curves of all populations of each model.

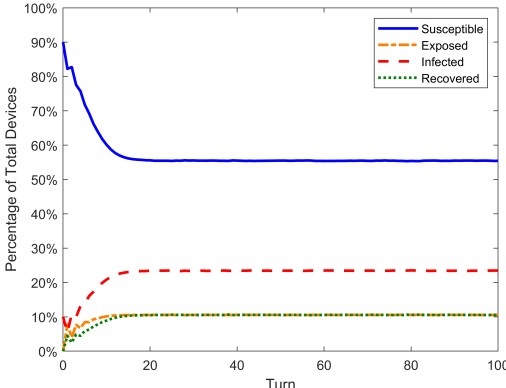

**Figure 8.** Results of simulation of models SEIRS for $\alpha = 90/\beta = 45$, with the curves of all populations of each model.

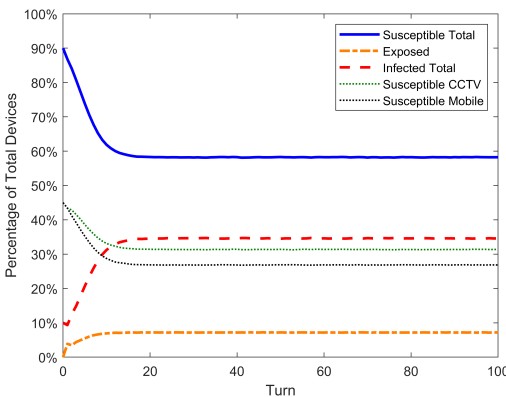

**Figure 9.** Results of simulation of models 2SEIS for $\alpha = 90/\beta = 45$, with the curves of all populations of each model.

**Table 6.** Comparison of performance between the standard nonlinear model and our proposed model of random propagation, applied in the epidemic models SEIRS and 2SEIS. Errors "Max E" and "RMSE" are calculated comparing the results of a mathematical model with its equivalent simulation.

| DTMC Model | | SIS | | | SEIRS | | | 2SEIS | | |
|---|---|---|---|---|---|---|---|---|---|---|
| Value of $\alpha$ | | Low | Average | High | Low | Average | High | Low | Average | High |
| Max Error of S | Standard | 3.15% | 5.96% | 9.88% | 2.54% | 4.05% | 5.58% | 2.95% | 5.32% | 8.34 |
| | Proposed | 0.14% | 0.13% | 0.20% | 0.13% | 0.13% | 0.14% | 0.11% | 0.16% | 0.11% |
| RMSE of S | Standard | 2.69% | 5.44% | 9.33% | 2.10% | 3.68% | 5.23% | 2.53% | 4.83% | 7.88% |
| | Proposed | 0.0510% | 0.0551% | 0.0582% | 0.0652% | 0.0551% | 0.0504% | 0.0480% | 0.0716% | 0.0532% |
| S[$T_f$] | Standard | 50.0% | 50.0% | 50.0% | 50.0% | 50.0% | 50.0% | 50.0% | 50.0% | 50.0% |
| | Proposed | 53.0% | 55.7% | 59.6% | 52.4% | 53.9% | 55.46% | 52.8% | 55.2% | 58.2% |
| $T_c$ of S | Standard | 36 | 18 | 11 | 39 | 22 | 13 | 38 | 21 | 13 |
| | Proposed | 35 | 18 | 10 | 39 | 21 | 13 | 38 | 20 | 12 |

### 6.1. Parameters

To nullify the variability of the stochastic simulation, a network model of $10^6$ IoT devices was used, at which size the variability of our random tests decreases to $\sim$0.1%. The network is isolated from external interactions, and the simulations start with a percentage of devices already infected from the beginning, which then propagate the malware to other devices. This initial percentage of infections is $\frac{I[0]}{N} = 10\%$ in all simulations.

Two important parameters are the rate of infection $\alpha$ and detection $\beta$. Their ratio approximates $R_0$, the "basic reproduction number", which defines the behavior of the malware epidemic: $R_0 < 1$ creates a scenario of natural eradication of malware; $R_0 = 1$ creates a balance between infection and mitigation, leading to slow transitions; and $R_0 > 1$ propagates malware aggressively until an endemic state with high percentages of infection.

$$R_0 = \frac{\alpha}{\beta} \tag{54}$$

The focus of our previous work was a mitigation strategy that compensated weak security in IoT networks, modeled as low detection rates of malware by the detection system. This goal motivated our choice of parameters in a different way when compared to the present study. We validated our propagation model for three different scenarios in SIS dynamics: $R_0 = 0.5$, $R_0 = 1$, and $R_0 = 2$, which defines the behavior of the malware epidemic as described above. We used the same low rate of infection $\alpha = 25\%$ to emulate weak network security and varied $\beta$ to generate each of these three $R_0$ values according to Equation (54). With this choice of parameters, our 2021 study [32] proposed a network-level mitigation strategy for constrained networks, while our 2023 study [21] extended it to global networks.

In this work, we focused on the generalization of derivation of the propagation dynamics and on the investigation of its improvement over the standard nonlinear model.

One of the main objectives was to identify the maximum errors of both models. Since the previous work limited $\alpha$ to 25%, this time we varied it to three values representing weak, average, and aggressive rates of infection: a **low** value of $\alpha = 25\%$, an **average** value of $\alpha = 50\%$ and a **high** value of $\alpha = 90\%$. In the previous study, we identified that bigger errors occur when $R_0 > 1$ (aggressive rate of infection); therefore, this time, we fixed $R_0$ at $R_0 = 2$ and made $\beta = \alpha/2$ for each of the three chosen values for $\alpha$ described above.

Besides $\alpha$ and $\beta$, the SEIRS model has two additional parameters: the exposed delay $\omega$ and the recovery delay $\tau$. The same value was used for them in all results: $\omega = 1$ and $\tau = 1$. Although bigger delays are important to explore the behavior of the SEIRS model, the objective here is to investigate the difference in precision between the two propagation models (our proposed model and the standard nonlinear model). Since the measured variable is the infected population $I$, the most accentuated difference of precision will happen when the infected population reaches its highest value. However, when the delays are big, devices stay longer in the delay-related state, inflating their size and consequently decreasing the size of $S$ and $I$. Therefore, the delays were kept at the very minimum to still have SEIRS behavior and create the biggest difference in precision between propagation models.

The 2SEIS model has only the infection delay but requires the previous definition of how many devices are CCTV and how many are mobile. 50% of the network was set for each. Regarding $E'$, the same value of exposed delay $\omega = 1$ was used.

### 6.2. Validation of Proposed Propagation Terms for SEIRS and 2SEIS

Table 6 presents four metrics to evaluate the accuracy of prediction of our proposed DTMC and the standard nonlinear one. The four metrics were calculated only for the curve of susceptible devices $S$, which in all simulated scenarios has the biggest value of any population. Consequently, it displays the biggest percentage error. "Max Error of S" and "RMSE of S" compare the results of a DTMC with its equivalent simulation (SIS, SEIRS, or 2SEIS). The first metric is simply the maximum error between DTMC and simulation. The second one represents the root-mean-square error of the time series of $S$ between DTMC and simulation. Its calculation is presented in Equation (55). "$S[T_f]$" represents the final value of the susceptible population of DTMC. At last, "$T_c$ of S" represents the time constant of the $S$ time series. "Time constant" here is defined as how many turns it takes for $I[T]$ to reach 95% of its final value.

$$RMSE = \sqrt{\frac{\sum_{k=1}^{T_{max}} (S_{sim}[k] - S_{dtmc}[k])^2}{T_{max}}} \tag{55}$$

In addition to the table results, the time series of $S$ is shown for both propagation models and for the simulation in Figures 10–12, as well as the time series of error between DTMCs and simulation in Figures 13–15. Like the figures of the last subsection, results are shown only for the scenario of high $\alpha$.

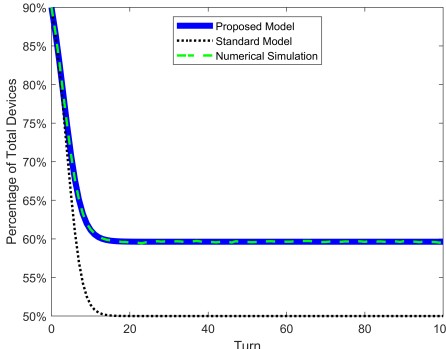

**Figure 10.** SIS epidemic model. Curve of population S for proposed model, standard nonlinear model, and simulation.

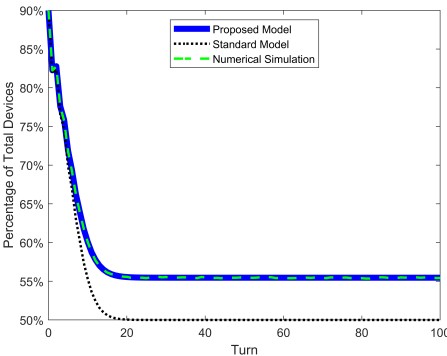

**Figure 11.** SEIRS epidemic model. Curve of population S for proposed model, standard nonlinear model, and simulation.

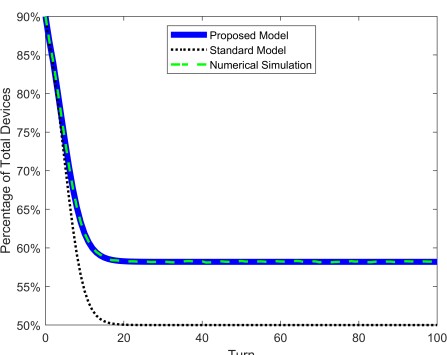

**Figure 12.** Double susceptible–exposed–infected–susceptible (2SEIS) epidemic model. Curve of population (S + S') for proposed model, standard nonlinear model, and simulation.

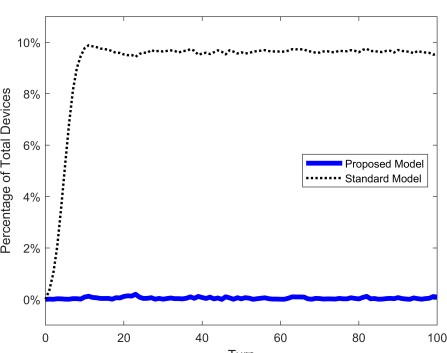

**Figure 13.** SIS epidemic model. Error of population S for proposed model, standard nonlinear model, and simulation.

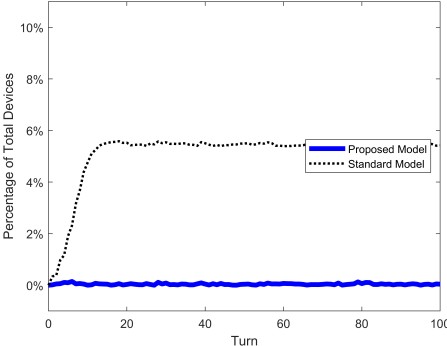

**Figure 14.** SEIRS epidemic model. Error of population S for proposed model, standard nonlinear model, and simulation.

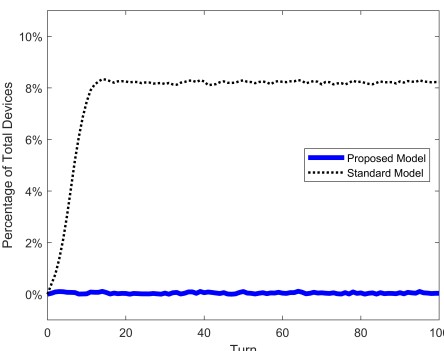

**Figure 15.** Double susceptible–exposed–infected–susceptible (2SEIS) epidemic model. Error of population (S + S') for proposed model, standard nonlinear model, and simulation.

The first result to discuss is the accuracy of our proposed model and the inadequacy of the standard nonlinear model. The Max Error of $S$ in the latter ranges $\sim$3% for low $\alpha$, $\sim$5% for average $\alpha$ and $\sim$8% for high $\alpha$, with a maximum error over all cases of 9.88%. In our previous work, the maximum error was only 3.15% for the SIS model (mainly because a new mitigation strategy focused on low infection and detection rates was investigated). But it turns out that the error of the standard nonlinear model can go as high as $\sim$10%. On the other hand, our proposed model has the same range of error of $\sim$0.15% over all scenarios and models. This confirms our previous conclusion: such a small and constant error is just the variability of the stochastic simulation and is independent of parameters and models. The same conclusion can be taken from Figures 13–15, where the error of $S$ for the standard model increases as the turn passes and the infections grow, while the error for our proposed model oscillates very slightly around 0%. Moreover, the RMSE of our model is extremely low ($\sim$0.05%); meanwhile, the standard nonlinear model has RMSEs as high as 9.33%. These quantitative results clearly demonstrate that our model is an exact match for the phenomenon of random propagation, independently of parameters or epidemic models.

Conceptually, such differences between propagation models can be explained by actual dynamics the standard nonlinear model represent exactly: pseudo-random propagation dynamics with only one inefficiency mechanism, the malware targeting of already-infected devices. The repetition of targets is not included in the small expression of $\alpha SI$. More details of this conclusion can be found in the previous work, where they demonstrated this exact match for the standard model with simulations.

The two last metrics showed the qualitative differences between the two compared propagation models. $S[T_f]$ of the standard model is 50% for all scenarios, but in our exact model, it is higher than 50% and increases with $\alpha$. This is an important conceptual difference, especially in light of our mitigation strategy proposed in the previous work. It has a trade-off between the number of nondetected infections being cleaned in the group mitigation and the unnecessary cleaning (generating operation downtime) of already clean devices. Our control policy is then based on calculating the number of blind cleanings to match how much percent of the network is infected. The reason for the difference in $S[T_f]$ between standard and proposed models can be understood by checking the dynamics of $S$ in the equilibrium state for both models. Taking the 2SEIS example, the expression for $S[T_f]$ is given in Equation (56). It is independent of $I$, being simply the inverse of $R_0$ times $N$.

$$0 = -\alpha S[T_f] \frac{I[T_f] + I'[T_f]}{N} + \beta(I[T_f] + I'[T_f])$$

$$S_{2seis/std}[T_f] = \frac{\beta}{\alpha} N = \frac{\beta}{R_0} \tag{56}$$

$S[T_f]$ of our proposed model, however, depends not only on $I[T_f]$ but also on the binomial (Equation (57)). Together with the numerical simulation, this mathematical result points to the overestimation of malware propagation in the standard model.

$$0 = -S[T_f]\left(1 - \left(1 - \frac{\alpha}{N}\right)^{I[T_f]+I'[T_f]}\right) + \beta I[T_f]$$

$$S_{2seis}[T_f] = \frac{\beta I[T_f]}{\left(1 - \left(1 - \frac{\alpha}{N}\right)^{I[T_f]+I'[T_f]}\right)} \tag{57}$$

The last metric ($T_c$) does not show any significant difference between these two models (however, it does when compared with the linear model, as shown in our last work). It is presented here anyway to add the small observation of how many turns it takes for the malware epidemic to stabilize for different ranges of $\alpha$. Considering how $\alpha$ almost doubled in every change of scenario, $T_c$ of $S$ almost shows a linear dependence. However, investigation of this point is left in future studies.

These results show that the standard nonlinear model, widespread in the literature, is not a good model for true random propagation of malware, and that our proposed model represents the exact dynamics of random propagation with the small trade-off of a slightly more complex expression (a binomial with six operations against a multiplication with three operations). Being both models nonlinear, the advantages of using the standard model instead of ours seem very small.

## 7. Conclusions

In this paper, we presented the generalization of our exact model of random propagation of malware to any discrete malware model that makes Markov chain assumptions. We further developed an alternative methodology of derivation that applies two rules to modify the simplest form of the propagation dynamics (made for the SIS model) to any compartmental malware model, simplifying the derivation process.

To validate this generalization, we performed the two forms of derivation for three malware models—SIS, SEIRS, and 2SEIS—evaluated the equations with varying sets of parameters and compared their results with their corresponding simulation. By comparing the time-series progression of the network populations between the results of equations and simulations, we found stochastic errors of less than 0.2%. This comparison was repeated for the standard nonlinear model (the most popular model in the literature), finding an average error of 5% and maximum error of 9.88% (almost two orders of magnitude).

This result confirms that our model is exact, and that errors in the standard nonlinear model can have a significant impact on predictability. Therefore, we highlight the advantage of substituting the standard nonlinear model with our proposed model. The only limitation to its use is the requirement of Markov chain assumptions, preventing its application in malware models with memory of the states $S$ and $I$ (models with memory of other states can still use our propagation dynamics, as seen in SEIRS).

In future works, we will present an investigation of the algebraic tractability of our propagation dynamics the in laws of control of automatic mitigation systems.

**Author Contributions:** Conceptualization, R.M.C. and Y.F.; Formal analysis, R.M.C. Funding acquisition, Y.F. and J.S.; Investigation, R.M.C. and Y.L.; Methodology, R.M.C.; Project administration, J.S.; Resources, Y.F. and J.S.; Software, R.M.C. and Y.L.; Supervision, Y.F. and J.S.; Validation, R.M.C.; Writing—original draft, R.M.C.; Writing—review and editing, R.M.C., Y.F. and J.S. All authors have read and agreed to the published version of the manuscript.

**Funding:** This research was conducted under a contract of "Research and development on IoT malware removal/make it nonfunctional technologies for effective use of the radio spectrum" among "Research and Development for Expansion of Radio Wave Resources (JPJ000254)", which was supported by the Ministry of Internal Affairs and Communications, Japan.

**Data Availability Statement:** https://github.com/rodrigo-carnier/exact-malware-propagation (accessed on 3 February 2024).

**Conflicts of Interest:** The authors declare no conflicts of interest.

## Abbreviations

The following abbreviations are used in this manuscript:

| | |
|---|---|
| 2SEIS | Double susceptible–exposed–infected–susceptible |
| CCTV | Closed-circuit television |
| DDoS | Distributed denial of service |
| DTMC | Discrete-time Markov chain |
| IoT | Internet of Things |
| RMSE | Root-mean-square error |
| SEIRS | Susceptible–exposed–infected–recovered-susceptible |
| SEIRD | Susceptible–exposed–infected–recovered–dead |
| SIR | Susceptible–infected–recovered |
| SIRS-L | Susceptible–infected–recovered–susceptible–low-energy |
| SIS | Susceptible–infected–susceptible |
| WSN | Wireless sensor network |

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
