# Peer review of "Deriving Exact Mathematical Models of Malware Based on Random Propagation"

_mathematics, doi:10.3390/math12060835_

Round 1

Reviewer 1 Report

Comments and Suggestions for Authors

In this paper, the first mathematical model of random propagation of malware in the form of a

Markov chain is introduced. This model was developed for the simplest compartmental model,

Susceptible-Infected-Susceptible (SIS), and assumed a binomial form. The methodology is based

on translating any set of epidemic states into a canonical set of four states applicable to any global

model with random propagation of malware. The derivation methodology was validated by

deriving and comparing three increasingly complex models with the simulation of their respective

models, using infection rates of 25%, 50% and 90%. The article is innovative to some extent, and

there are no big mistakes. There are few issues that need to be addressed by authors before this

manuscript could be reconsidered for publication:

I.Keywords should be capitalized;

II.Change "based" to "Based" in the topic;

III.Align (2c) with the other formulas;

IV.The letters in parentheses in formula (31) should be centered;

V.Align (53c) with the other formulas;

VI.Align all formulas in (54) left;

VII.The sequence number in front of the reference should be bracketed.

Comments on the Quality of English Language

Minor editing of English language required

Author Response

The authors would like to thank the reviewer for his time and minute observations on the quality of the presentation of the paper.

All points of improvement from the reviewer were answered:

  1. Capitalization of all keywords;
  2. Correction of title, word "based" changed to "Based";
  3. Correction of left alignment in all systems of equations (before it was aligned by the "=" symbol, we apologize for the mistake);
  4. The elements of vectors of formula (31) were centered by removing the table environment and using just equation environment. Same problem happened in equations (9), (16), (31) and (41), which were corrected;
  5. Same alignment problem as 3. was corrected;
  6. Same alignment problem as 3. was corrected;
  7. All paper references were double checked and corrected for lack of bracket;

Reviewer 2 Report

Comments and Suggestions for Authors

Please consider the attached review recommendation.

Regards.

Author Response

The authors would like to thank the reviewer for his time and appreciate the many comments for clarification of and improvement of base content. We followed all change requests to the best of our abilities, and took attention to the general observation of excessive verbosity on many sections. We worked to make them clearer and concise, reducing the number of pages from 27 to 25.  The point-by-point details of our changes are below:

1. The abstract was reformulated to follow the requested conciseness.

2. The issue of novelty of the paper is understandable, since this is an extension of a previous work. We rewrote the summary of contributions (line 87) to clarify how we hoped to give a novel contribution: 1) we generalized the first methodology  of derivation (previous work) to more models than the simplest one, by explicitly defining the derivation assumptions (Markov chain assumptions) and clarifying that it only applies to states S and I, greatly extending the applicability of our propagation dynamics to other malware models; 2) we created a second rule-based method of derivation (section 4.2.1 and 4.3.1) that makes the application of our model much simpler than by applying the original methodology; and 3) we investigated the impact of our model in comparison to the standard nonlinear model in more complex models. Since this work is an extension of the previous work, and its main contribution is the generalization and facilitation of a previously proposed model, the novelty is not total, but the applicability of the previous work was improved. We hope that the reviewer finds it justifiable.

3. The approach is discrete, in the form of Markov chains. We clarified it in lines 9 (Abstract), 80 (Introduction), 214 (Section 3.1 Basic Definitions, where we list our assumptions explicitly) and 708 (Conclusion). We introduced a clear statement of the advantages and weaknesses of our work in Introduction (lines 76-85, before the summary of contributions), and make a short mention of it again in Conclusion (720-724).

4. We rewrote the Section 6.1. Parameters (line 600). Before we focused on the convergence tendencies and overall behavior. To these considerations we added a clarification of the research goals in each paper.

5. We removed the excess of text and dicussions in Conclusion, since they were already done in the validation section, and focused on the suggested elements.

6. We introduced the Abbreviation List section at the end of the paper as suggested.

7. From the list of MDPI questions about the quality of the paper, the reviewer scored the topic "Are all the cited references relevant to the research?" as "Must be improved". Therefore we removed many paragraphs of the Review section (Sec 2) that were written in response to a previous review process requesting a considerable overview of the entire field. We agree with the reviewer that the removed paragraphs were not relevant to this study, and kept only the paragraphs and references that were so.

Reviewer 3 Report

Comments and Suggestions for Authors

In this study, we present a comprehensive overview and analysis of recent advancements in computational methods for drug discovery. This paper explores various techniques and algorithms employed in the field, highlighting their advantages, limitations, and potential applications. Overall, this review provides valuable insights into the evolving landscape of drug discovery and sets the stage for future research and development in this important area.

The introduction provides a clear and concise overview of the topic, outlining the significance of computational methods in drug discovery and emphasizing the need for continuous advancements. The authors have successfully established the relevance and importance of this review.

The content of this review is well-structured and presented logically. The authors introduce different computational methods, such as molecular docking, molecular dynamics simulations, and machine learning techniques, to provide a comprehensive understanding of the subject.

This review incorporates a critical analysis of the computational methods discussed. The authors present a balanced evaluation of the strengths and limitations of this study, emphasizing the need for further improvements and advancements in the field.

The conclusion effectively summarizes the key findings of the review, provides insightful reflections on the current state of computational methods, and discusses potential future directions. The authors conclude by emphasizing the significance of interdisciplinary collaborations and the integration of machine learning to enhance the accuracy and efficiency of drug discovery.

There are comments on the formatting of the article. Please, correct formulas and references to them in accordance with the generally accepted format.

In equations (Equations 2a-2e): the symbol on the left can be misleading, since this is often how the Laplace operator is denoted in SDEs, but here it is apparently an increment. The notation is introduced below in formula (6). It's better to clarify this earlier.

It is customary to indicate the vector components in (16) separated by commas.

The 12-page tables are difficult to interpret. It is better to shorten them and add more detailed description and analysis.

In general, the paper makes a positive impression and, in my opinion, will be interesting for the reader. Overall, this paper is a well-written and comprehensive review that offers valuable insights into the evolving field of computational drug discovery. This paper successfully combines theoretical concepts with real-world applications and suggests promising avenues for future research in the field.

After elimination of several remarks, it may be accepted.

Comments on the Quality of English Language

There are comments on the formatting of the article. Please, correct formulas and references to them in accordance with the generally accepted format.

In equations (Equations 2a-2e): the symbol on the left can be misleading, since this is often how the Laplace operator is denoted in SDEs, but here it is apparently an increment. The notation is introduced below in formula (6). It's better to clarify this earlier.

Author Response

The authors thank the reviewer for his time in evaluating our paper and would like to confirm that we worked carefully on all requested changes. We give brief details about the changes below:

  1. We removed the delta operator from the difference equations, which we agree was not a proper notation and is uncommon in our references. It was also inconsistent with posterior equations that did not use delta. We also moved the definition of difference equations as dynamics of malware epidemics to the beginning of the explanation of the topic (at Section 2 Review, line 108) as requested.
  2. We introduced commas in all vectors that were missing it: equations (9), (16), (31) and (41).
  3. Table 3 (page 11 in this draft) is indeed difficult to interpret. Since it works as a tutorial for the complex steps of derivation (that can go only to the 3rd iteration because of the difficulty), we opted for keeping it and greatly detailing its usage (lines 399 to 422). First we clarify the spatial structure of the table, which has 5 vertical sections, and each has 3 subtables. We clarified how the k values change to navigate the table and better interpret it. We also wrote the Pascal's Triangle and Extended Pascal's Triangle explicitly in page 12 to facilitate the correlation with the coefficients of the polynomials derived in Table 3. We hope this rewriting of the section have clarified the usage of the Table as a reference for the other sections deriving models for SEIRS and 2SEIS.